

# FoBSim: an extensible open-source simulation tool for integrated fog-blockchain systems

Hamza Baniata and Attila Kertesz

Department of Software Engineering, University of Szeged, Szeged, Hungary

## ABSTRACT

A lot of hard work and years of research are still needed for developing successful Blockchain (BC) applications. Although it is not yet standardized, BC technology was proven as to be an enhancement factor for security, decentralization, and reliability, leading to be successfully implemented in cryptocurrency industries. Fog computing (FC) is one of the recently emerged paradigms that needs to be improved to serve Internet of Things (IoT) environments of the future. As hundreds of projects, ideas, and systems were proposed, one can find a great R&D potential for integrating BC and FC technologies. Examples of organizations contributing to the R&D of these two technologies, and their integration, include Linux, IBM, Google, Microsoft, and others. To validate an integrated Fog-Blockchain protocol or method implementation, before the deployment phase, a suitable and accurate simulation environment is needed. Such validation should save a great deal of costs and efforts on researchers and companies adopting this integration. Current available simulation environments facilitate Fog simulation, or BC simulation, but not both. In this paper, we introduce a Fog-Blockchain simulator, namely FoBSim, with the main goal to ease the experimentation and validation of integrated Fog-Blockchain approaches. According to our proposed workflow of simulation, we implement different Consensus Algorithms (CA), different deployment options of the BC in the FC architecture, and different functionalities of the BC in the simulation. Furthermore, technical details and algorithms on the simulated integration are provided. We validate FoBSim by describing the technologies used within FoBSim, highlighting FoBSim's novelty compared to the state-of-the-art, discussing the event validity in FoBSim, and providing a clear walk-through validation. Finally, we simulate case studies, then present and analyze the obtained results, where deploying the BC network in the fog layer shows enhanced efficiency in terms of total run time and total storage cost.

## INTRODUCTION

In light of the general tendency towards skepticism around Blockchain (BC) systems being reliable, huge research and industrial projects are being encouraged to address issues and vulnerabilities of those systems. This is because it is believed that a successful BC deployment would definitely advance Internet-of-Everything (IoE) applications. Dubai,

Corresponding author
Hamza Baniata,
baniatah@inf.u-szeged.hu

for example, has planned for being the first smart city powered by BC (*Smart Dubai Department, 2020*). China had launched, in late 2019, a BC-based smart city ID system (*Global Times, 2019*), while it is planning to have its own official digital currency (*Smartcity Press, 2019*). Before that, Liberstad, a private smart city in Norway, has officially adopted City Coin as its official currency (https://www.liberstad.com/).

BC is a Distributed Ledger Technology (DLT) in the form of a distributed transactional database, secured by cryptography, and governed by a consensus mechanism (*Beck et al., 2017*). This technology was first introduced as the backbone of the Bitcoin ecosystem in 2009 (*Bitcoin.org, 2009*). As BC got high reputation and attention among research and industry communities, as well as governments, it has proven robustness against the disadvantages of classical centralized systems. Furthermore, different versions, uses, paradigms, and platforms were proposed, aiming to extend the deployment of BC beyond cash and payment purposes.

Concerning smart things, homes, and cities, Fog Computing (FC) paradigms become reality. FC is a horizontal, physical or virtual resource paradigm that resides between smart end-devices and traditional cloud data centers (*Markakis et al., 2017*). FC is conceptually an extension of the cloud at the edge of the network. Hence, most cloud services should be introduced by the fog layer as well, except the fog provides better latency measures.

Different reference architectures were proposed for the FC paradigm, e.g., by *Habibi et al. (2020)*, *Dastjerdi et al. (2016)*, *OpenFog Consortium (2017)*, and Cisco (*Bonomi et al., 2014*). Nevertheless, they all have the same general properties of middling between end-users and the clouds, providing cloud services at the edge of the network, managing mobility issues, and introducing reliable and secure communications.

We have previously investigated the integration of BC with FC in *Baniata & Kertesz (2020)*. Accordingly, we concluded that such integration may ease the optimization of several current Cloud-Edge issues, such as enhancing security, credibility, and resource efficiency. Also, decentralizing FC applications decreases the appearance of single points of failure and the control of a centralized authority. However, we found that major challenges still need more research efforts such as:

- The lack of individual standardization of both technologies, FC and BC, which leads to the lack of standardization of the integration of them.
- Many privacy issues and threats remain, such as the location awareness property of fog components, which raises some concerns.
- Ironically, as FC enhances the latency of end-user applications, BC causes the exact opposite, if the consensus mechanisms are not properly designed. Other major issues may also represent barriers if this latency issue was not addressed, such as authentication, scalability, and heterogeneity problems. This is because solving the latency problem may require waiving some advantageous protocols or mechanisms of FC.

- The aforementioned challenges may further lead to somewhat low trust levels of the BC-FC integration, which is the main cause of the illegalization of BC technologies in general.

Consequently, the research and industry communities have been working hand-in-hand to solve these major challenges, along with other technical issues. Such efforts require reliable and flexible simulation environments that can mimic real-life scenarios with the lowest possible costs. Old, out-dated, or somewhat close simulation tools that were initially implemented for classical Peer-to-Peer networks, such as PeerSim (*Montresor & Jelasity, 2009*), may not be able to cover all the mechanisms of a modern BC system. Although some recently proposed systems use PeerSim, such as *Petri et al. (2020)*, it surly required vast amount of changes, modifications, and additions to redesign it into a BC simulation tool.

In this paper, we propose a Fog-Blockchain simulation environment, called FoBSim, that is able to simulate different integration scenarios of FC and BC. Concerning our main contributions, we discuss and analyze the architectural elements of FC- and BC-based systems, and present the modules, algorithms, and strategies implemented in FoBSim. We also describe in detail the validation, the incentivization, and the confirmation mechanisms deployed in the current version of FoBSim. To exemplify its utilization, we discuss possible application scenarios of FC-BC integration, and we clarify how such applications can be simulated and optimized using FoBSim. The abbreviations we use within our paper are declared in Table 1, while the main properties of the current version of FoBSim are as follows:

1. FoBSim provides different Consensus Algorithms (CA), namely Proof-of-Work (PoW), Proof-of-Stake (PoS) and Proof-of-Authority (PoA) that are ready to be deployed in any scenario.
2. FoBSim facilitates the deployment of BC miners in the fog or end-user layer.
3. FoBSim allows different services to be reliably provided by the BC network, namely Data Management, Identity Management, Computational Services (through Smart Contracts (SC)), and Payment/Currency transfer Services.
4. FoBSim provides both, parallel execution and non-parallel execution, of mining processing. While gossiping is optionally and efficiently available so that the distributed chain is consistent in different possible network topologies.
5. FoBSim is the first simulation environment whose primary goal is to mimic integration scenarios of FC and BC technologies.

The remainder of the paper is organized as follows: "Related Work" presents and discusses state-of-the-art simulation environments that are maybe suitable to simulate FC-BC systems. To properly introduce FoBSim, we discuss, in detail, how FC architectural elements are deployed in "FC Architectural Elements". Additionally, we discuss the categories of BC systems, each with its properties and components in "BC Architectural Elements". Accordingly, we propose the components, the algorithms, and the functions

**Table 1 Description of abbreviations used within the manuscript.**

| Abbreviation | Description | Abbreviation | Description |
| --- | --- | --- | --- |
| BC | Blockchain | PoW | Proof of Work |
| FC | Fog Computing | PoS | Proof of Stake |
| IoT | Internet of Things | PoET | Proof of Elapsed Time |
| CA | Consensus Algorithm | PoA | Proof of Authority |
| IoE | Internet-of-Everything | TTP | Trusted Third Party |
| DLT | Distributed Ledger Technology | P2P | Peer-to-Peer |
| SC | Smart Contracts | TX | Transaction |
| GUI | Graphical User Interface | TTL | Time To Live |
| QoB | Quality of Blockchain | DAG | Directed Acyclic Graph |
| PoG | Proof of Generation | MT | Merkle Tree |

of the FoBSim environment in "The Fobsim Environment". To validate FoBSim, we simulate some use cases and present the simulation results in "Case Studies". Finally, we present our future work and conclude in "Conclusions".

# RELATED WORK

Searching the literature for tools specifically implemented for simulating FC-BC integration scenarios, we found that no previous work has directly targeted our objective. That is, we found several simulation tools that mimic fog-enhanced cloud systems, IoT-Fog-Cloud scenarios, etc., and several tools that mimic BC scenarios, each with specific constraints on the used CAs. Nevertheless, some proposals for IoT-BC simulation tools can be somewhat related to our work. For example, the ABSOLUT tool, investigated in *Kreku et al. (2017)*, models the deployment of BCs in IoT environments. Accordingly, some critical analysis were provided regarding network latency, effects of miners number on the overall efficiency of the IoT network, and simulation errors.

*Liaskos, Anand & Alimohammadi (2020)* proposed a general architecture that a BC simulation needs to follow in order to be considered comprehensive. Further, some properties were declared as necessary for encouraging the adoption and re-usability of the simulation. The proposed architecture includes extensible connection strategies, BC nodes, BC chains, Transactions (TXs) and Transaction pools, users, events, Blocks, and most importantly Consensus mechanisms. Events can include different triggers to other events—that may be performed by any entity of the network—(such as TX/block arrival, TX/block validation, connection requests, etc.). Also, Events need to be handled by concise and well implemented strategies.

In light of the lack of simulation tools similar to our proposal, we found it more suitable to present this section in two separate groups: namely FC simulation tools, and BC simulation tools.

## FC simulation tools

Recently, our research group has started to investigate the state-of-the-art related to cloud, IoT and fog simulation tools in *Markus & Kertesz (2020)*. Within this study, several

simulation tools were classified, compared, and analyzed, such as the DockerSim tool (*Nikdel, Gao & Neville, 2017*), FogNetSim++ (*Qayyum et al., 2018*), and EdgeCloudSim (*Sonmez, Ozgovde & Ersoy, 2018*). Furthermore, technical details, advantages, vulnerabilities, and software quality issues were also discussed.

*Rahman et al. (2019)* surveyed 15 simulation tools for cloud and data centers networks scenarios. The tools were discussed and compared according to several criteria, such as the Graphical User Interface (GUI) availability, the language with which the simulator was implemented, and the communications model. Consequently, they proposed the Nutshell tool which addresses some drawbacks that were ignored by most of the surveyed simulators. For example, most surveyed simulators had abstract network implementation and low-level details were missing. Further, non of the studied tools provided an addressing scheme, a congestion control mechanism, or a traffic pattern recognition mechanism. Out of those 15 presented simulation tools, seven were defined as extensions of the CloudSim toolkit (*Calheiros et al., 2011*).

*Yousefpour et al. (2019)* presented a complete survey about FC, referencing 450 publications specifically concerned with FC development and applications. Within their extended survey, some FC simulation tools, such as iFogSim (*Gupta et al., 2017*; *Naas et al., 2018*), Emufog (*Mayer et al., 2017*), Fogbed (*Coutinho et al., 2018*), and MyiFogSim (*Lopes et al., 2017*) were discussed. As iFogSim was conceptually built using the CloudSim communications model, it inherited some of its properties, such as the ability to co-execute multiple tasks at the same time and the availability of plugable resource management policies.

Generally speaking, any cloud simulation tool can be extended to be a fog-enabled simulation tool. This is because of the fundamental property of the fog layer acting as a bridge between end-users and the cloud. In other words, adding a fog module to a cloud simulation tool, describing communications, roles, services, and parameters of fog nodes, is sufficient to claim that the tool is a fog-enhanced cloud simulation tool. Additionally, in a project that targets a Fog-BC integration applications, many researchers used a reliable, general-purpose fog simulator and implemented the BC as if it was an application case, such as in *Kumar et al. (2020)*. The results of such simulation approach can be trusted valid for limited cases, such as providing a proof of concept of the proposal. However, critical issues, such as scalability and heterogeneity in huge networks, need to be simulated in a more specialized simulation environments. To mention one critical case, the BC protocols deployed in different CAs require more precise and accurate deployment of the BC entities and inter-operation in different layers of a Fog-enhanced IoT-Cloud paradigm. Consequently, as some simulation scenarios need an event-driven implementation, while others need a data-driven implementation, a scenario's outputs may differ when simulated using different simulation environments. Such possibility of fluctuated simulation outputs should normally lead to unreliable simulation results.

## BC simulation tools
As we have previously investigated how a Fog-Blockchain integration is envisioned, we started the implementation of FoBSim with a simple BC simulation tool described in

*Baniata (2020)*. Consequently, we discuss the state of the art regarding BC simulation tools available in the literature. In later sections, we describe how FoBSim serves as a reliable tool to mimic an FC-BC integration scenario.

*Anilkumar et al. (2019)* have compared different available simulation platforms specifically mimicking the Ethereum BC, namely Remix Ethereum (*Ethereum, 2020*), Truffle Suite (*Truffle Blockchain Group, 2020*), Mist (*Bahga & Madisetti, 2017*), and Geth (*Bruno, 2018*). The comparison includes some guidelines and properties such as the initialization and the ease of deployment. The authors concluded that Truffle Suite is ideal for testing and development, Remix is ideal for compilation and error detection and correction, while Mist and Geth are relatively easy to deploy. *Alharby & Van Moorsel (2019)* and *Faria & Correia (2019)* proposed a somewhat limited simulation tool, namely BlockSim, implemented in Python, which specifically deploys the PoW algorithm to mimic the Bitcoin and Ethereum systems. Similarly, *Wang et al. (2018)* proposed a simulation model to evaluate what is named Quality of Blockchain (QoB). The proposed model targets only the PoW-based systems aiming to evaluate the effect on changing different parameters of the simulated scenarios on the QoB. For example, average block size, number of TXs per block/day, the size of the memPool, etc. affecting the latency measurements. Furthermore, the authors identified five main characteristics that must be available in any BC simulation tool, namely the ability to scale through time, broadcast and multi-cast messages through the network, be Event-Driven, so that miners can act on received messages while working on other BC-related tasks, process messages in parallel, and handle concurrency issues.

*Gervais et al. (2016)* analyzed some of the probable attacks and vulnerabilities of PoW-based BCs through emulating the conditions in such systems. Sub-consequently, they categorized the parameters affecting the emulation into consensus-related, such as block distribution time, mining power, and the distribution of the miners, and network-related parameters, such as the block size distribution, the number of reachable network nodes, and the distribution of those nodes. They basically presented a quantitative framework to objectively compare PoW-based BCs rather than providing a general-purpose simulation tool.

*Memon et al. (2018)* simulated the mining process in PoW-based BCs using the Queuing Theory, aiming to provide statistics on those, and similar systems. *Zhao, Guo & Chan (2020)* simulated a BC system for specifically validating their proposed Proof-of-Generation (PoG) algorithm. Hence, the implementation objective was comparing the PoG with other CAs such as PoW and PoS. Another limited BC implementation was proposed by *Piriou & Dumas (2018)*, where only the blocks appending and broadcasting aspects are considered. The tool was implemented using Python, and it aimed at performing Monte Carlo simulations to obtain probabilistic results on consistency and the ability to discard double-spending attacks of BC protocols. In *Deshpande, Nasirifard & Jacobsen (2018)*, the eVIBES simulation was presented, which is a configurable simulation framework for gaining empirical insights into the dynamic properties of PoW-based Ethereum BCs. However, the PoW computations are excluded in eVIBES, and the last updates on the code were committed in 2018.

**Table 2 Blockchain simulation tools and their properties.**

| Refs. | PL | PoW | PoS | PoA | SC | DM | PM | IDM | F |
|---|---|---|---|---|---|---|---|---|---|
| *Alharby & Van Moorsel (2019)* and *Faria & Correia (2019)* | Python | ✓ | χ | χ | ✓ | χ | ✓ | χ | χ |
| *Wang et al. (2018)* | Python | ✓ | χ | χ | χ | χ | ✓ | χ | χ |
| *Memon et al. (2018)* | Java | ✓ | χ | χ | ✓ | χ | χ | χ | χ |
| *Zhao, Guo & Chan (2020)* | Python | ✓ | ✓ | χ | χ | ✓ | χ | χ | χ |
| *Piriou & Dumas (2018)* | Python | χ | χ | χ | χ | χ | ✓ | χ | χ |
| *Deshpande, Nasirifard & Jacobsen (2018)* | Java | ✓ | χ | χ | ✓ | χ | ✓ | χ | χ |
| FoBSim | Python | ✓ | ✓ | ✓ | ✓ | ✓ | ✓ | ✓ | ✓ |

To highlight the comparison between the mentioned BC simulation tools and our proposed FoBSim tool, we gathered the differences in Table 2. PL, PoW, PoS, PoA, SC, DM, PM, IDM, and F are abbreviations for Programming Language, Proof-of-Work, Proof-of-Stake, Proof-of-Authority, Smart Contracts, Data Management, Payment Management, Identity Management, and Fog-enhanced, respectively. As shown in the table, none of the previously proposed BC simulation tools makes the PoA algorithm available for simulation scenarios, provides a suitable simulation environment for identity management applications, or, most importantly, facilitates the integration of FC in a BC application.

Many other references can be found in the literature, in which a part of a BC system, or a specific mechanism is implemented. The simulated "part" is only used to analyze a specific property in strict conditions, or to validate a proposed technique or mechanism under named and biased circumstances, such as in *Wang et al. (2020)* and *Raman et al. (2019)*. It is also worth mentioning here that some open-source BC projects are available and can be used to simulate BC scenarios. For example, the HyperLedger (*The Linux Foundation, 2020*) projects administered by the Linux Foundation are highly sophisticated and well-implemented BC systems. One can locally clone any project that suits the application needs and construct a local network. However, those projects are not targeting the simulation purposes as much as providing realized BC services for the industrial projects. Additionally, most of these projects, such as Indy, are hard to re-configure and, if re-configured, very sensitive to small changes in their code. Indy, for example, uses specifically a modified version of PBFT CA, namely Plenum, while Fabric uses RAFT.

## FC ARCHITECTURAL ELEMENTS

The FC layer can be studied in three levels, namely the node level, the system level, and the service level (*Farhadi et al., 2020*). The fog consists of several nodes connected to each other and to the cloud. The main purpose of the fog layer is to provide cloud services, when possible, closer to end-users. Further, the fog layer, conceptually, provides enhanced security and latency measures. Hence, an FC system uses its components in the fog layer to provide the services that end-users request from the cloud.

In a simple scenario, the fog receives a service request from end-users, performs the required tasks in the most efficient method available, and sends the results back to

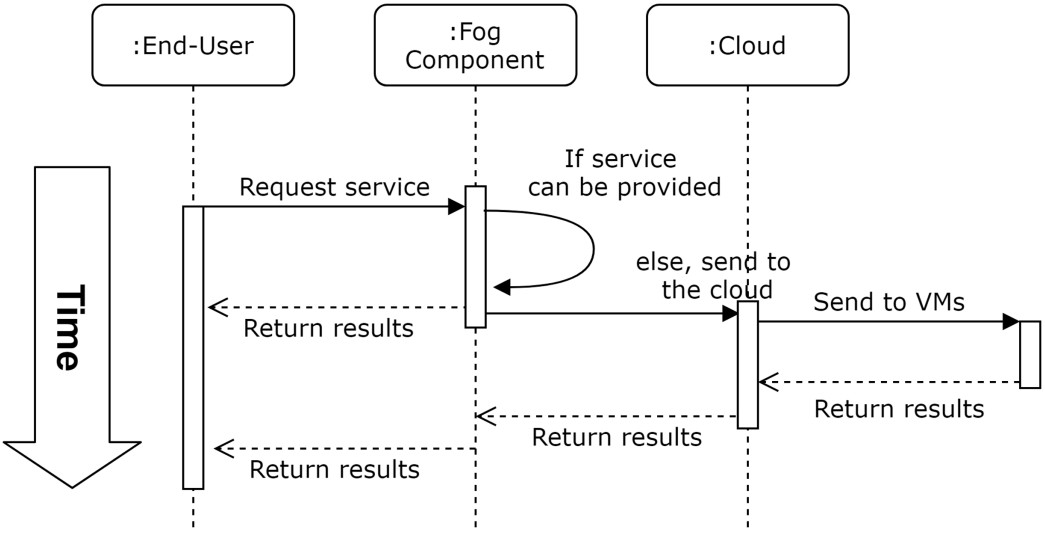

**Figure 1 Workflow of an automated fog-enhanced cloud system.**

end-users. As the clouds mainly provide Infrastructure, Software, and Platform-as-a-Service models, those three models can be used for computational tasks, storage tasks, or communication tasks (*Maes et al., 2018*).

For a fog-enhanced cloud system, a general overview of the workflow is presented in Fig. 1. As presented in the figure, the service is requested from end-users and the fog layer provides this service if possible, otherwise, the request is forwarded to the cloud where complex and time consuming actions are performed. However, information of the complexity of the system, and the decision making process in the fog layer, should not be within the concern of end-users. That is, end-users require their tasks to be performed within a privacy-aware context and the QoS measures implications that were agreed on.

In FoBSim, the fog layer can be configured according to the scenario that needs to be simulated. For example, the number of fog nodes, the communications within the fog layer and with other entities of the simulated system, and the services provided by the fog, can all be modified.

## BC ARCHITECTURAL ELEMENTS

BC is a DLT that consists of several elements which need to efficiently interact with each other, in order to achieve the goal of the system. A general view of BC systems suggests some fundamental components that need to be present in any BC system. A BC system implies end-users who request certain types of services from a BC network. The BC network consists of multiple nodes, who do not trust each other, that perform the requested services in a decentralized environment. Consequently, the service provided by a BC network can only be valid if the BC network deployed a trusted method, i.e., CAs, to validate the services provided by its untrusted entities.

In FoBSim, the BC network can provide two models of services; namely data storage, and computations. Meanwhile, the communications within the BC network and with the

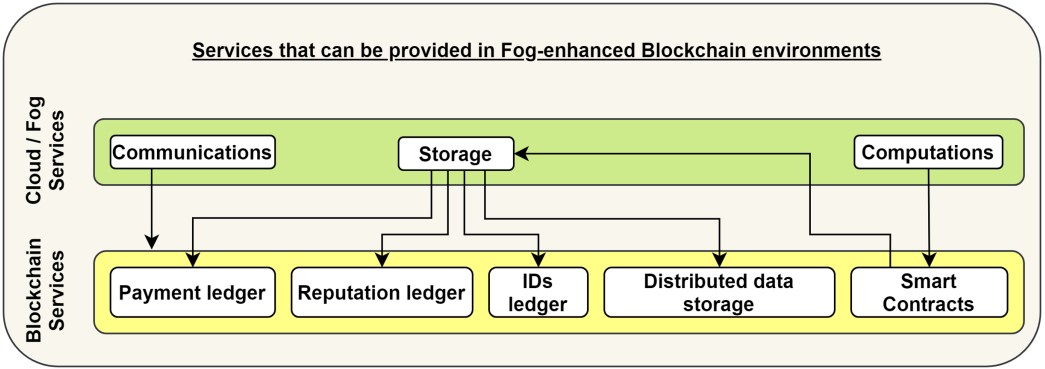

**Figure 2 Service models provided by cloud/fog systems, and their relevant service models provided by BC systems.**

fog layer are configurable. Data storage service model implies that pieces of data are saved on the immutable distributed ledger. Such data may be of any type including data records, IDs, digital payment registration, or reputation measures of end-users or fog components. It can also be noted that some applications require assets to be transferred between clients, such as cryptocurrency transfer applications or real estate ownership applications. Other applications do not require transferring assets rather than saving data on the chain only, such as voting applications and eHealth applications. However, the mentioned second type of applications may also need, on some level, a digital payment method be embedded. In such cases, SCs on other payment platforms can be implemented and generated, such as Bitcoin or Ethereum.

Performing computations for end-users is the second service model that the BC in FoBSim can be configured to provide. That is, computational tasks can be sent by end-users/fog entities to the BC in the form of SC, which are small chunks of code, run by BC nodes upon fulfillment of algorithmically verifiable conditions (*Coladangelo & Sattath, 2020*). After running the SCs, the results can be saved in a centralized or decentralized form according to the pre-run configuration. Figure 2 presents how the services, classically provided by a cloud/fog system, can be interpreted into the form of services that can be provided by a BC system. We can notice in the figure that SCs can be considered relevant to cloud computational services, while different types of data saved on the decentralized BC can be considered a relevant option to the centralized storage model provided by a cloud system.

## Consensus algorithms

Several approaches were proposed as a solution for the aforementioned needs, among which are the most famous PoW CA. PoW was deployed in 2009 in the first BC system, i. e., Bitcoin (*Nakamoto, 2019*), and is currently used in other robust BC systems; such as Ethereum (*Vujičic, Jagodić & Ranđić, 2018*). Although PoW methods have proven strong security and support to BC systems, they have some drawbacks, such as high energy consumption and high latency, that encouraged the R&D communities to search for other trusted methods.

The PoS algorithm (*King & Nadal, 2012*) was proposed a couple of years later in order to solve the high energy consumption problem implied by PoW. PoS is currently being optimized to provide similar advantages as PoW. Ethereum, for example, is planning to substitute PoW with PoS in the very near future. However, some drawbacks of PoS need to be solved before its official deployment, such as The Monopoly Problem (*Larimer, 2013*), The Bribe Attack (*Bentov, Gabizon & Mizrahi, 2016*; *Deirmentzoglou, Papakyriakopoulos & Patsakis, 2019*), and relatively low reliability (*Zhang & Chan, 2020*).

In PoW-based BCs, a BC node proves the validity of its generated block of data by coupling a puzzle solution within the block. The puzzle solution is generally characterized by hardship to be obtained while it can easily be validated once found. Generally, the puzzle is a mathematical problem that requires high computational power to be obtained. In PoS-based BCs, the BC node that is allowed to generate the next block is chosen randomly by the system. To encourage the system to pick a specific BC node, staking more digital coins in deposit shall increase the probability of being chosen. This provides high trust measures as faulty generated blocks are not tolerated by the system, and the staked coins of the malicious/faulty BC node would be burned as a penalty.

Other approaches were proposed that provide trust in BCs. Examples include the PoET (*Buntinx, 2017*), and the PoA (*Avasthi & Saxena, 2018*). PoET-based BCs generate randomly selected times for BC nodes. The one node whose randomly picked time elapses first, is the one who is granted the opportunity to generate the next block. PoA, on the other hand, implies that only blocks signed by authorized members are validated and confirmed by the BC network. Those authorized nodes must be known trusted participants that can be tracked and penalized in case of faulty behavior. Both of these CAs share the property of being suitable for private and permissioned BCs, while PoW and PoS are known for being suitable for public and permissionless BCs.

FoBSim allows to choose the suitable CA according to the simulated scenario. While there are many versions of each CA mentioned, we currently provide the simplest version of each so that modifications can be performed with no complexities. To obtain more information about them, more details can be found at *Sheikh (2018)*, *Singh et al. (2019)*, and *Chen et al. (2017)*.

### Transactions

In a very simple scenario, an end-user sends a request to the BC network, which consists of BC nodes, to perform a defined TX. As stated at the beginning of this section, TXs may be data to be stored (i.e., payment data, reputation data, identity data, etc.), or can be SCs whose results can be either saved in a centralized (in the case of Cloud) or distributed manner (in the cases of fog or BC). Once the TX is performed, it should be agreed on by the majority of BC nodes if to be saved on the distributed ledger and, sub-consequently, be added to the chain saved in all BC nodes.

On the other hand, if the fog layer is controlling and automating the communications between the end-user layer and the BC network, as in *Baniata & Kertész (2020)*, the TXs are sent from end-users to the fog. After that, some communication takes place

between the fog layer and the BC network in order to successfully perform the tasks requested by end-users. In such system model, we assume that the BC network lays in a different layer than the fog layer. The case where the BC network is placed in the fog layer is covered in "Functionality of the BC Deployment". Nevertheless, a feedback with the appropriate result of each TX should be easily achievable by end-users.

## Distributed ledger

In the case were data needs to be stored in a decentralized manner, no Trusted Third Party (TTP) needs to be included in the storing process. The entity considered as a TTP in regular fog-enhanced cloud systems is the cloud, where data is stored. However, computations can take place in the fog layer to enhance the QoS.

Within DLT-enabled systems, such as BC, groups of data are accumulated in blocks, and coupled with a proof of validity, as explained in "Consensus Algorithms". Once a new block of TXs is generated, and the proof is coupled with them, the new block is broadcast among all BC nodes. Nodes who receive the new blocks verify the proof and the data within each TX, and if everything is confirmed valid, the new block is added to the local chain. With each BC node behaving this way, the new block is added to the chain in a distributed manner. That is, a copy of the same chain, with the same exact order of blocks, exists in each BC node. Further, a hash of the previous block is added to the new block, so that any alteration attack of this block in the future will be impractical, and hence almost impossible.

## Functionality of the BC deployment

As a BC-assisted FC system can provide computational and storage services, the BC placement within the FC architecture may differ. That is, BC can be placed in the fog layer, the end-user layer, or the cloud layer. In FoBSim, however, we consider only the first two mentioned placement cases.

When the BC is deployed in the fog layer, storage and computational services are performed by the fog nodes themselves. In other words, fog nodes wear a second hat, which is a BC network hat. Thus, when storage to be provided by the fog while fog nodes are also BC nodes, data is stored in *all* fog nodes in the fog layer. A simple system model is demonstrated in Fig. 3A, where only one chain is constructed in the lower fog layer and one fog control point in the upper layer monitors the BC functionality. However, such a model is not practical and more complexities appear in a real-life scenario, including heterogeneous fog nodes, multiple BC deployments, different CAs, and different service models. In such complex systems, FoBSim can be easily extended by adding the needed classes and modules and, hence, cover necessary proposed scenario entities. A note is worth underlining here is the importance of differentiating between the services provided by fog nodes which are BC nodes, and the services provided by fog nodes which are not BC nodes. The first type gets incentivized by end-users for providing both fog services and BC services, while the second type gets incentivized by end-users for providing only fog services. Such critical issues need to be taken care of, when simulating Fog-BC scenarios, to maximize the reliability of the obtained results.

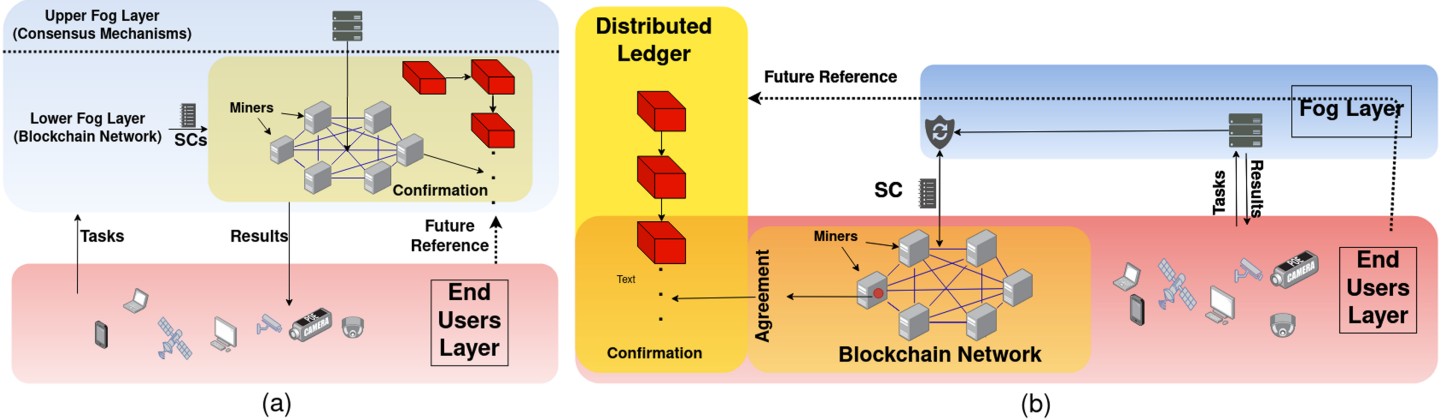

**Figure 3** FC-BC integration system model, where (A) the BC is deployed in the fog layer, and (B) the BC is deployed in the end-user layer.

In a system model where the BC is deployed in the end-user layer, we can distinguish two types of end-users; namely task requester and BC node. In a fog-enhanced BC system, the fog controls the communication between the two types of end-users. Specifically, BC nodes perform the tasks that were sent to the BC network by the fog, which originally were requested by task requester end-users. Further, the fog can control the privacy preservation of data and incentivize BC nodes in the form of digital currency, as in *Baniata, Anaqreh & Kertesz (2021)*. To be specific, BC nodes can be further sub-categorized according to the scenario to be simulated. Adding other types of BC nodes is up to the developers and the system model. For example, the Bitcoin system is modeled in a simpler way, where BC is directly connected to task requester end-users, and it only provides a payment ledger service. Ethereum, on the other hand, provides computational and data management services. This makes Ethereum surpass Bitcoin because it can provide more services to end-users. FoBSim improves both system models by optionally adding the fog layer. The system model provided by FoBSim when the BC is deployed in the end-user layer is demonstrated in Fig. 3B.

## THE FOBSIM ENVIRONMENT

To cover all architectural elements described in "FC Architectural Elements" and "BC Architectural Elements", we implemented FoBSim according to the conceptual workflow demonstrated in Fig. 4. The current version of FoBSim covers all the architectural elements of a BC system and an FC system. This means that FoBSim successfully inlines with the general architecture of a reliable BC simulation presented in *Liaskos, Anand & Alimohammadi (2020)*. In fact, many more services and scenarios can be simulated using FoBSim, covering the fog layer inclusion besides the BC. As presented in Fig. 4, different CAs can be used, different services of the BC network can be declared, and different placement scenarios of the BC network can be chosen. When the BC network is located in the fog layer, the number of BC nodes does not need to be input because, as described earlier, each fog node is also a BC node. Nevertheless, the number of task requester end-

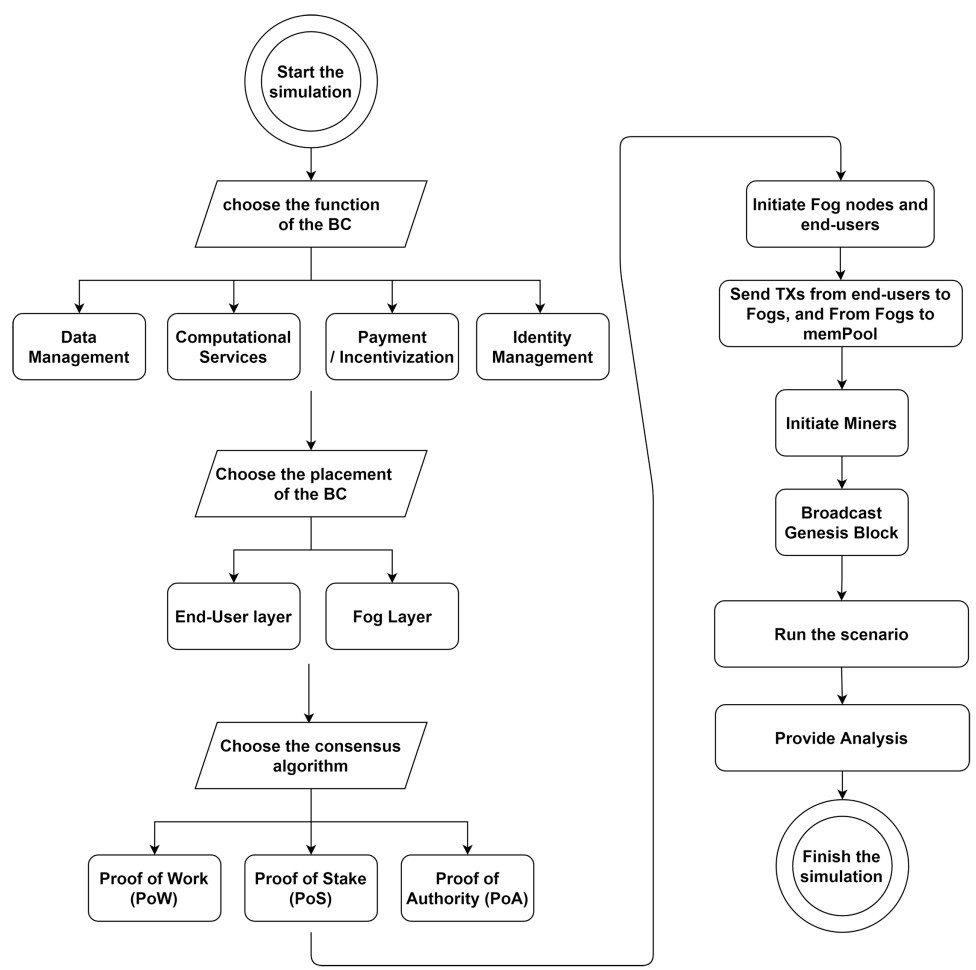

**Figure 4 Workflow of a simulation run using the FoBSim environment.**

users connected to each fog node needs to be input, while some fog nodes in a PoA-based scenario might not be authorized to mint new blocks. Once the network is built, running and testing the system model can take place.

The FoBSim environment is implemented using Python v3.8, with the inclusion of some common packages such as: *random*, *randrange*, *multiprocessing*, *time*, and *hashlib*. The current version of FoBSim can be cloned and directly run as all the variables, lists, dictionaries, and sets have been given initial values. However, these parameters can be modified before running the code in the *Sim_parameters.json* file. FoBSim tool is open-source and freely available at *Baniata & Kertesz (2020)*.

## FoBSim modules

To facilitate the understanding of FoBSim, we demonstrate the methods within each FoBSim module in Fig. 5. Further, we show the classes and methods of FoBSim modules

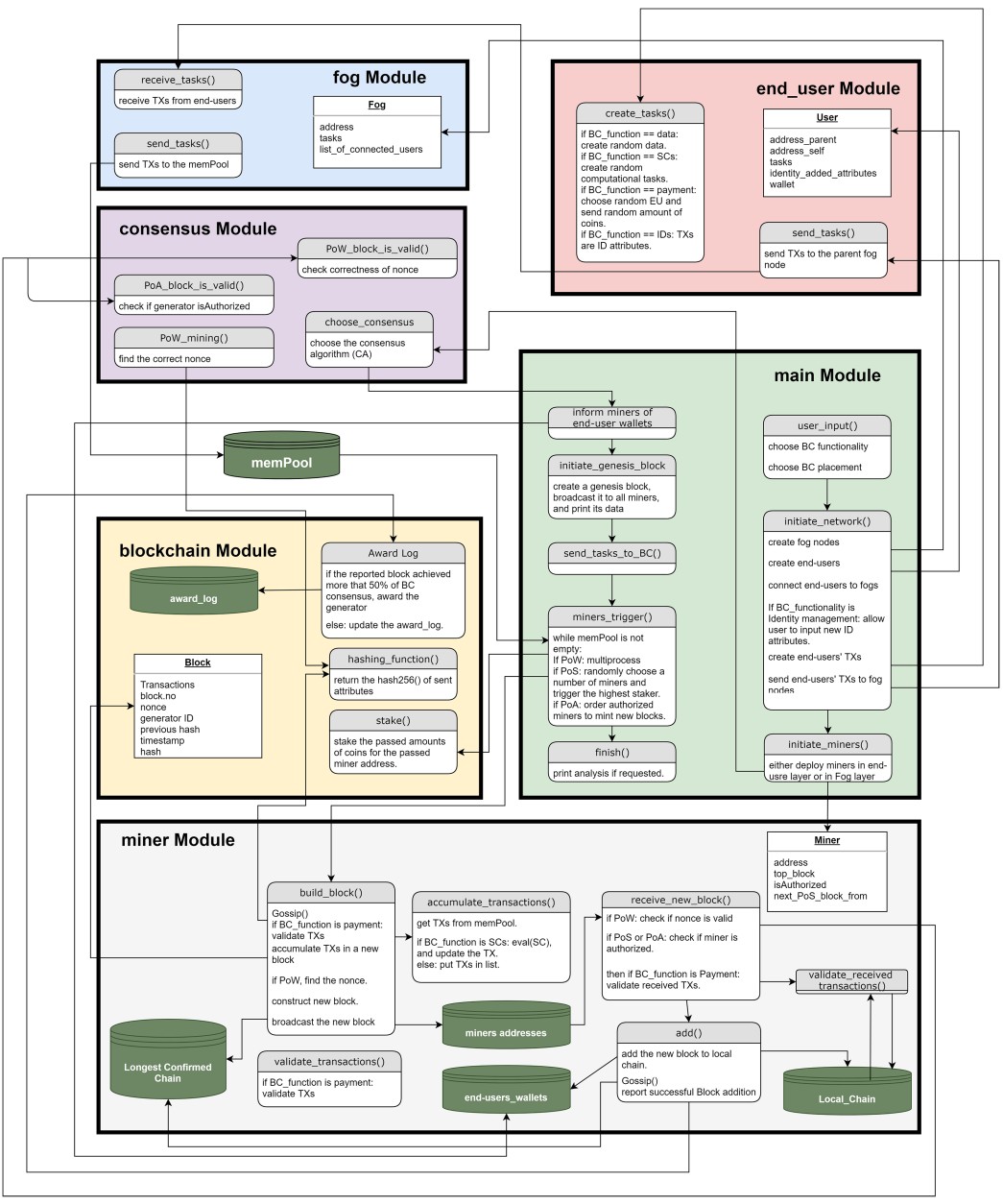

**Figure 5 The interaction among modules and methods of the FoBSim environment.**

in tables of the Supplemental Material for this paper. Some notes to be taken care of need to be underlined as well:

1. There is a big opportunity for developers to implement new methods in the fog layer. For example, the fog nodes can be extensible to provide privacy-preserving mechanisms (such as described in *Baniata, Almobaideen & Kertesz (2020)*), computational services (such as described in *Fröhlich, Gelenbe & Nowak (2020)*), or reputation and trust management services (such as described in *Debe et al. (2019)*).

2. memPool.py: In this module, the mempool, where TXs are accumulated, is a Python multiprocessing queue that allows different processes to synchronously add() and get() TXs.

3. There are other minor methods from other modules also called by FoBSim entities that mint a new Block, or receive a new TX/Block, in order to synchronously and smoothly apply each different CA's policies, as declared in its simple version.

4. After each simulation run, some temporary files can be found in the temporary folder of FoBSim. These files are originally initiated by the main module, the BC module, or the miner module. The temporary files are used synchronously by different FoBSim entities, mimicking the real-world interaction between BC entities. The current version of FoBSim generates some or all of the following files depending on the simulated scenario:

- Miners' local chains.
- Miners' local records of users' wallets.
- Log of blocks confirmed by the majority of miners.
- Log of final amounts in miners' wallets (initial values − staked values + awards).
- Log of coin amounts which were staked by miners.
- The longest confirmed chain.
- Forking log

## Genesis block generation

The first block added to the chain in each simulation run is the most important block of the chain. Different scenarios imply different formats of this block, and different methods to broadcast it among, and be accepted by, miner nodes. In the current version of FoBSim, however, a genesis block is initiated with a list of TXs containing only the string "genesis_block" and the labels of the miners available when this block was generated. The block number is 0, the nonce is 0, the generator_id is "The Network", the previous hash is 0, and the hash is generated using the **hashing_function** in the blockchain.py module. The timestamp of the genesis block indicates when the chain was launched, hence all blocks shall have bigger timestamp values than the genesis timestamp. Figure 1 of the Supplemental Material of this paper shows a standard FoBSim genesis block, generated in a BC network that consists of two miner nodes.

## FoBSim consensus algorithms

Currently, there are three available CAs ready to be used in different simulation scenarios. Next, we describe each one individually as to facilitate any modifications by developers. However, we need to indicate that the three included CAs are in their simplest versions and may require some individual modification in case of the need of more sophisticated ones. Before delving into the CAs, however, we need to discuss the Gossip protocol in FoBSim, as it is deployed regardless of what CA is chosen.

### Gossip protocol

A Gossip Protocol (*Blywis et al., 2011*) is usually deployed in P2P systems for maintaining the consistency of distributed data saved in decentralized networks. Specifically in BC systems, miner nodes regularly, yet randomly, gossip to their neighbors about their current version of the chain, aiming to reach consensus finality as soon as possible. According to specific characteristics of the BC, the locally saved chains are updated so that all confirmed chains are equivalent at any given moment (*He, Cui & Jiang, 2019*). The equivalency that any BC system is seeking is defined by the contents similarity of the chains (i.e., TXs, hashes, etc.), and the order similarity of the confirmed blocks. That is, a chain $[b_1, b_2, b_3]$ is not equivalent to $[b_1, b_3, b_2]$ despite the fact that both have similar contents.

Gossiping protocols are usually fault tolerant as many failing nodes do not affect the protocol. Furthermore, they can adapt to the dynamics of the network, so some solutions have been proposed in the literature for nodes joining and leaving the network. However, gossiping is an iterative method that never quits as long as the network is up, and it may take time to converge. Additionally, a high level of communication costs is expected for gossiping, while randomly chosen neighbors are informed about updates. Thus, one cannot provide precise analysis about the time needed for the network agreement on a piece of data.

Although the implementation of such protocol is relatively simple, it is differently implemented in different systems. Some famous examples of efficient gossiping protocols include the Push-Sum protocol (*Kempe, Dobra & Gehrke, 2003*), the Push-Flow algorithm (*Gansterer et al., 2013*), and different versions of the Push-Pull averaging protocol (*Gabor & Jelasity, 2018*). Furthermore, we found that its application in FoBSim is useful, when the PoW CA is used in a multiprocessing scenario, with a relatively low puzzle difficulty. Additionally, it can be easily noted that the number of simulated TXs/blocks and the initial TX per block configuration affect the speed of the system to reach consensus finality. That is, for low numbers of TXs, blocks, and low ratios of TXs per block, miners might not have the required time to converge locally saved chains. Accordingly, final versions of local chains in some FoBSim simulations, under such circumstances, may not coincide, which is normal and expected as described in *Fan et al. (2020)*. Nevertheless, we deployed a simple Push-Pull Gossip version in FoBSim that works perfectly fine, so that modifications can be easily conducted if needed. In the current version of FoBSim, a Time To Live (TTL) parameter was not added to the Pull requests when gossiping. This, as expected, floods the network with Pull and Push requests each time a node wants to gossip. Nevertheless, we faced no problem whatsoever when the network consisted of up to 1,500 miners. If more miners need to be deployed in the simulation scenario, where gossiping is activated, we recommend either configuring the gossiping requests to have a TTL (i.e., a number of hops the requests perform before they are terminated), and/or decreasing the number of neighbors the gossiping node is sending the gossip request to. That is, instead of gossiping with all neighbors, a miner can randomly choose a neighbor to gossip with. Consequently, each neighbor will gossip with a randomly chosen neighbor of his, etc. More details on such implementation approach can be found in *Lan et al.*

---

**Algorithm 1** The default Gossip protocol in FoBSim.

**Result:** Confirmed Local_chain in $\mu_g$

**initialization:** Self(miner $\mu_g$);

confirmed_chain = self.local_chain;

temporary_global_chain = longest_chain;

Condition_1 = len(temporary_global_chain) > len(confirmed_chain);

Condition_2 =blocks in temporary_global_chain are confirmed by network majority;

**if** Condition_1 **AND** Condition_2 **then**

confirmed_chain = temporary_global_chain;

self.local_chain = confirmed_chain;

self.top_block = confirmed_chain[str(len(confirmed_chain)-1)];

**if** BC_function is Payment **then**

self.log_users_wallets = confirmed_chain_from.log_users_wallets

**end**

**end**

---

*(2003)*, while detailed analysis regarding the success rate of gossiping, with a given TTL in a given P2P network, can be found in *Bisnik & Abouzeid (2007)*.

Algorithm 1 describes how the Pull-request in the default Gossip protocol of the current version of FoBSim works. If the gossiping property was set to *true*, Each miner runs this algorithm each time the Gossip() function is called for that miner (as a default, the Gossip function is called each time a miner is triggered to build a new block and when a new block is received). As demonstrated in the algorithm, a default FoBSim miner requests information about the longest chain, and adopts it if its contents were agreed on by the majority of the network, which is a condition tested using Algorithm 2. Additionally, if a miner receives a new valid block, and the resulting local chain was longer than the global chain, the miner updates the global chain instantly, which represent the Push request of the Gossip protocol in FoBSim.

In big BC networks, the mentioned issues need to be carefully designed, so that the consistency of the distributed ledger by the end of the simulation run is guaranteed, while the efficiency of the algorithm is optimized.

### The proof of work

In a simplified scenario of a PoW-based BC, miners collect TXs from the mempool (which is a shared queue in FoBSim) and accumulate them in blocks that they mint. Specifically, all available miners *compete* to produce the next block that will be added to the chain. The fastest miner producing the next block is the miner whose block is accepted by all other miners of the BC. Synchronously, all blocks that are being minted by other miners are withdrawn, and all TXs within are sent back to the mempool. To mimic this

---

**Algorithm 2** The default chain confirmation function in FoBSim.

**Result:** bool **chain_is_confirmed**

**Passed parameters:** Chain **C**, network size;

**initialization:** chain_is_confirmed = True;

block_confirmation_log = blockchain.confirmation_log;

Condition_1 = **not** (C[block]['hash'] in block_confirmation_log);

Condition_2 = block_confirmation_log[chain[block]['hash']]['votes'] <= (network size / 2);

**for** *block in C* **do**

    **if** *Condition_1 OR Condition_2* **then**

        chain_is_confirmed = False;

        **break**

    **end**

**end**

return **chain_is_confirmed**

---

scenario in FoBSim, we needed to deploy the multiprocessing package of Python and trigger all miners to work together on the next block.

Each miner then works within an isolated core of the device on which the simulation is conducted. Using this approach is doable and explainable in simple scenarios, where each process needs to access one or few shared objects. However, we found it challenging to mimic complex scenarios, where huge number of processes require accessing the same shared lists. For example, when the BC functionality is payment, the BC is deployed in the fog layer, and the CA is PoS, the wallets of end-users, fog nodes, and mining nodes need to be all global for read and update by all processes. We also experimented the Python package: multiprocessing.shared_memory, which partially solved the problem as multi processes can read and update values in a Shareable List object. However, as declared in the official Python documentation (*The Python Software Foundation, 2020*), the Shareable List object lacks the dynamicity required in terms of length and slicing. According to the mentioned insights, we implemented two approaches for PoW mining in FoBSim, the first starts all miners in parallel (using the multiprocessing package), while the second consequentially calls for miners to mint new blocks (using a FOR loop). Both approaches are available in the **miners_trigger()** function in the main.py module, and developers are free to use either. We do encourage developers, however, to be cautious and carefully test their results when using the parallel processing approach, as each different scenario may require different access management scheme to different FoBSim entities. Hence, a complex scenario simulation may require some modifications to some variables and lists so that they become shareable by all processes in different modules. Detailed instructions for implementing different memory-sharing scenarios can be found in the Python official documentation (*The Python Software Foundation, 2020*).

When a miner receives a new block, it checks whether the hash of the block (in which the nonce or the puzzle solution is included) is in line with the acceptance condition

---

**Algorithm 3** The default PoW mining algorithm in FoBSim miner.

**Result:** New block β confirmation

**initialization** Self(miner μ*g*);

Collect TXs from memPool;

Gossip();

**if** *BC_function is Payment* **then**

|     validate collected TXs

**else**

|     **if** *BC_function is Computational Services* **then**

|        |     eval(TXs);

|        |     add the evaluation results to TXs

|     **end**

|     Accumulate TXs in a new BC block β;

|     Find the puzzle solution of β (nonce);

|     Broadcast β to neighbors;

**end**

**if** *New block β is received* **then**

|     Gossip();

|     **if** β *nonce is correct* then

|        **if** *BC_function is Payment* then

|        |     validate and confirm TXs in β

|        **end**

|        add block β to the local chain;

|        Broadcast β to neighbors;

|        report a successful block addition [β, μ*g*]

|     **end**

**end**

---

enforced by the blockchain.py module. Further, the receiver miner checks whether sender end-users have sufficient amounts of digital coins to perform the TXs (in the case of payment functionality). Unlike PoS and PoA, all miners work at the same time for achieving the next block. Hence, any miner is authorized to produce a block and there is no miner verification required. Algorithm 3 presents how PoW is implemented in FoBSim.

### The proof of stake

In a simplified version of PoS, miners stake different amounts of digital coins (which they temporarily are not allowed to claim) in the BC network. The network then randomly chooses a miner to mint the next block, with higher probability to be chosen for miners who stake more coins. Once a miner is chosen, it is the only one authorized to mint

and broadcast the next block. In case of faulty TXs/blocks, the minter loses its staked coins as a penalty, while in case of correct blocks, the minter is awarded some digital coins.

To mimic this in FoBSim, each miner is initiated with a specific amount of coins in its wallet. After that, a randomly generated number of coins (up to the amount of coins in its wallet) is staked by each miner. This way, every miner has a different probability to be chosen by the network. Next, the network randomly chooses, say 10% of the available, miners and picks the one with the highest stake. This chosen miner's address is immediately broadcast to all miners so that any block received from any other miner is rejected. Once the new block is received, it is validated and added to the local chain. Algorithm 4 presents how PoS is implemented in FoBSim.

Here, a very wide space is available for implementing reputation management schemes in FoBSim. Different scenarios and different applications require different parameters affecting entities' reputation. Further, adding other types of miners, end-users, or even fogs implies that different DBs can be suggested.

It is also worth mentioning here that we found it unnecessary to use the multiprocessing package because only one miner is working on the next block. Hence, no competition is implied in the PoS scenario.

### The proof of authority

In a simplified version of the PoA algorithm. only authorized network entities (by the network administrators) are illegible to mint new blocks. Regardless of the BC functionality, there is also no need to deploy the multiprocessing package for PoA-based scenarios as there is no competition as in PoS-based scenarios.

To mimic the PoA in FoBSim, we allow the user to declare which entities are authorized to mint new blocks. The declaration requested from the user appears in the case of BC deployment in the fog or end-user layer. That is, each fog node is administering a group of end-users, and providing communications (and probably computations) services to them. However, it is not necessary for each fog node in the fog layer to be a BC node as well, but it should be there as only a fog node. Authorized fog nodes then are wearing both hats, fog nodes and BC miners. When the BC is deployed in the end-user layer, authorized miners are responsible for minting new blocks and maintaining the distributed ledger. Meanwhile, unauthorized miners are only responsible for validating new blocks, received from their neighbors, and for maintaining the distributed ledger.

This approach allows for comfortably emulating a scenario where the BC in the fog layer and part of the fogs are included in the BC functionality. Notice that a fog node that is also a BC node performs all the required tasks in logical isolation. This means that a fog node that is administering a group of end-users has a buffer to save the end-users TXs, but it does not use these TXs to mint a new block. Rather, it sends these TXs to the mempool as required, and then, only if it was authorized, it collects TXs from the mempool. Notice also, that the mempool is a simple queue in FoBSim, yet it can be implemented for some scenarios to be a priority queue. Our implementation of isolating the services provided by a fog node that is also a BC miner facilitates the simulation of scenarios where TXs need to be processed according to their priority. For example, miner

**Algorithm 4** The default PoS mining algorithm in FoBSim.

**Result:** Confirmed new block β

**initialization** miners $\mu_{[0,1,\dots n]}$, miners.wallets, stake random no. of coins from each miner.;

**The Network:**;

**while** *mempool.qsize() > 0* **do**

> Randomly choose a predefined no. of miners;
>
> Choose the miner with the highest Stake value;
>
> Inform all miners of the ID of the next block generator $\mu_g$;

**end**

**The Miner:**;

**if** a new ID μg is received from the Network **then**

> **if** *MyAddress* == $\mu_g$ **then**
>
> > Collect TXs from memPool;
> >
> > **if** *BC_function is Payment* **then**
> >
> > > validate collected TXs
> >
> > **else**
> >
> > > **if** *BC_function is Computational Services* **then**
> > >
> > > > eval(TXs);
> > > >
> > > > add the evaluation results to TXs
> > >
> > > **end**
> >
> > **end**
> >
> > Accumulate TXs in a new BC block β;
> >
> > Broadcast β;
>
> **else**
>
> > wait for a new block from $\mu_g$;
> >
> > **if** b is received **then**
> >
> > > **if** $\mu_g$ == β.*generator* **then**
> > >
> > > > **if** BC function is Payment **then**
> > > >
> > > > > validate and confirm TXs in β
> > > >
> > > > **end**
> > > >
> > > > add block β to the local chain;
> > > >
> > > > broadcast β to neighbors;
> > > >
> > > > report a successful block addition [β, $\mu_g$]
> > >
> > > **end**
> >
> > **end**
>
> **end**

**end**

nodes in Ethereum usually choose the SCs with the highest Gas/award provided by end-users. This is a type of prioritizing that can be simulated in FoBSim. Similarly, in Bitcoin, a priority value is computed for each TX according to Eq. (1), and TXs with higher fees and higher priority values are processed faster (*Narayanan et al., 2016*). The default PoA algorithm implemented in FoBSim is clarified in Algorithm 5.

$$Priority = \frac{\sum inputAge * inputValue}{TXsize} \tag{1}$$

## Transaction/block validation in FoBSim

Here, we need to underline some differences between the terms verification, validation, and confirmation, and we need to see how FoBSim differentiates between those terms in different scenarios. As we have touched on these differences in *Baniata & Kertész (2020)*, we need to accurately define each of these terms in order to correctly describe how FoBSim works.

Validation is the process when a miner (either a minter or receiver) checks the correctness of a claim. That is, in the case of a minter miner, the puzzle solution (or nonce) provided with the minted block needs to be correct before the block is broadcast. If the nonce was valid, the block is broadcast, otherwise, a new solution is searched for. While in the case of a receiver miner, the nonce is checked once. If in this later case the solution was valid, the block is accepted, otherwise, the block is rejected.

In the case of payment functionality, the validity of TXs fetched from the mempool is tested. This means that the amount of coins in the wallet of the sender of each TX, in the payment functionality, is compared to the amount to be transferred. If the wallet contains less than the transferred amount, the TX is withdrawn from the block. Later when the new block is received by a miner, the same hash validation and TXs validation take place, except if one of the TXs were invalid, the whole block is rejected. In the case of a block rejection, the minter miner is usually reported in a reputation-aware context. If all the contents of a newly received block are valid (i.e., the hash, the TXs, the wallets, the block number, and the nonce) the block is added to the locally saved chain. Here, we can say that TXs are confirmed, because the block is added to the chain (i.e. the block is confirmed).

The verification, on the other hand, is the process of verifying the identity of an entity. For example, in the case of PoA, only authorized miners are allowed to mint new blocks. Similarly, in the case of PoS, a received block should be generated by a miner that all other miners expect to receive the new block from. Additionally, public information about end-users' wallets need to be accessible by miners to validate their TXs. Thus, a received block, with some TXs generated by end-users who do not have wallets, or whose wallets contents are not readable by miners, can not be validated and confirmed. Failing to confirm a TX is not necessarily caused by end-users not having sufficient coins to transfer, but may also happen for end-users who can not be *verified*.

All of these critical principles are, by default, taken care of in FoBSim. All miners are informed about the end-users public identities and wallets' contents. After that,

---

**Algorithm 5** The default PoA mining algorithm in FoBSim.

---

**Result:** Confirmed new block $\beta$

**initialization** Fog nodes $\Psi_{[0,1,\ldots n]}$;

**if** *BC placement is Fog Layer* **then**

    User Input("address of authorized fog nodes")

**else**

    Input("address of authorized miners")

**end**

save authorized miners $\mu_{[0,1,\ldots n]}$ in Miners_List;

**The Miner:**;

**while** *mempool.qsize() > 0* **do**

    **if** *self.address* $\in \mu_{list}$ **then**

        collect TXs from memPool;

        **if** *BC function is Payment* **then**

            validate collected TXs

        **else**

            **if** *BC function is Computational Services* **then**

                eval(TXs);

                add the evaluation results to TXs

            **end**

        **end**

        accumulate TXs in a new BC block $\beta$;

        broadcast $\beta$ to neighbors;

    **end**

**end**

**if** $\beta$ *is received* **then**

    **if** $\mu_g \in \mu_{list}$ **then**

        **if** *BC function is Payment* **then**

            validate and confirm TXs in $\beta$

        **end**

        add block $\beta$ to the local chain;

        broadcast $\beta$ to neighbors;

        report a successful block addition [$\beta$, $\mu_g$]

    **end**

**end**

---

transferred coins are updated locally in each miner. Consequently, a new TX from the same end-user will be compared to the updated amount of coins in its wallet. Invalid TXs are not included in the block being minted, while invalid TXs cause the rejection of the

whole received block. Once a block's contents are validated, and the TXs/block generators are verified, the TXs are confirmed, the locally saved wallets amounts are updated, and the block is locally confirmed and added to the chain. The most interesting thing is that the very small probability of a double spend attack (*Karame, Androulaki & Capkun, 2012*) which can appear in PoW-based scenarios, can be easily simulated in FoBSim. All processes are actually happening during each simulation run, rather than substituting them with a small delay as in most BC simulation tools we checked. Hence, validation, verification, and confirmation processes can be modified according to the scenario to be simulated. Nevertheless, Bitcoin decreases the double spend attack probability by regularly raising the difficulty of the puzzle, which is a property that can be modified in FoBSim as well. To facilitate the simulation of such critical scenarios, we deployed two broadcasting approaches for newly minted blocks. The first allows the broadcast process using a simple FOR loop, where miners sequentially validate and confirm new blocks. The second allows the broadcast process using the multiprocessing package, which allows all miners to receive and process new blocks at the same time. Relatively, developers need to be cautious when using the second approach, because of some critical challenges similar to those mentioned in "The Proof of Work".

## Awarding winning miners

Generally speaking, BC miners get rewarded by two system entities for providing the BC service (i.e., BC functionality). The first is the end-user who generated the TX, who pays a small fee once the TX is confirmed (e.g., Gas in Ethereum). The second is the BC network itself (i.e., all miner nodes), which updates the winning miner's wallet once a new block (minted by the winning miner) is confirmed. We can notice here how important it is to clarify the difference between validation, verification, and confirmation. That is, a miner is verified by its public label and public wallet key/address (ID). Then, a miner being authorized to mint a new block is validated (claim). Finally, a miner is awarded for minting a conformable block (miner's wallet is updated).

In FoBSim, we implemented the second mechanism, where miners get rewarded for their services by the network. We assume this part is hard because it, also, needs to be agreed on by the majority of BC miners (i.e., at least 51%), and it requires the condition that they confirm the block. The default implementation of FoBSim does that. For the first incentivization mechanism, we thought that it is not applicable in many different scenarios, hence we left it for the developers to add it if needed. For example, to allow end-users to provide fees for getting tasks in the BC, one field can be added to generated TXs, containing the amount of fees the end-user is willing to pay for the service. Once a miner picks a TX (mostly, TXs with higher fees are faster to be picked and processed by miners) and the block containing the TX is confirmed, all miners add the TX fees to the winning miner's wallet. Figure 2A of the Supplemental Material presents a screenshot of FoBSim output, concluding that a new block was received from Miner_2 by Miner_3, and that the BC module just obtained the needed confirmations to consider the new block confirmed by the whole BC network. Thus, the minter is awarded. Later, the receiver miner presents its updated local chain according to the successful network

confirmation. On the other hand, Figure 2B of the Supplemental Material presents a screenshot of the miner_wallets_log after a simulation run, where the PoA CA was used and all miners, except for Miner_5, were authorized to mint new blocks (initial wallet value was 1,000).

## Strategies in FoBSim

As had been discussed so far, there are some default strategies used by FoBSim entities throughout each simulation run. To mention some, TXs are picked by miners with no preference, e.g., the highest Gas or priority. Also, a default chain is a single linear chain and new blocks are added to the top of this chain. Some applications, however, have multiple chains or multi-dimentional chains, e.g., Directed Acyclic Graph (DAG) based chains. Additionally, if two blocks appear in the network, the block that was accepted by the majority of miners is confirmed rather than, in some BC systems, the older one is confirmed even if it was confirmed by the minority. Further, a valid block is immediately announced, once found, into the FoBSim network, while in some applications, there might be a conditional delay. For instance, if a selfish mining attack scenario is to be simulated, miners would prefer to keep their newly found blocks secret, hoping they will find the next block as well (*Negy, Rizun & Sirer, 2020*).

The current version of FoBSim supposes that the data flows from end-users to the fog, and from the fog to the BC network. However, there are other possible data flow schemes that can be simulated, as depicted in Fig. 6. For example, the BC in the current version provides DLT services to end-users, which are communicating with the BC through the fog layer, while services might be provided by the fog layer to the BC network or from the BC network to the fogs in some applications. Further, an application where end-users may need to request data directly from the BC might be possible, which implies different data flow scheme as well. FoBSim facilitates the modification of the data flow in the simulated application, and presents an extra Cloud module that can add more possibilities to the application.

Network connectivity characteristics are a major and critical concern in any BC system. To facilitate network architects job, FoBSim allows to define the number of nodes in each layer, the number of neighbors of each BC node, and the general topology of the network. Additionally, all BC nodes are connected into one giant component by default, whether they were deployed in the fog layer or end-user layer. Accordingly, the effect of manipulating the topology of simulated networks can be easily captured.

## FoBSim constraints

Some properties have not been implemented in the current version of FoBSim, such as MT, Digital Signatures and Mining Pools. Additionally, FoBSim source code can be run on a PC with Microsoft Windows or Linux OS, but it may need some modifications if to be run on a PC with a MAC OS (some functions require access to OS operations such as deleting or modifying files located at the secondary memory). Finally, the default limit of recursion in Python may restrict the number of miners to 1,500, which may raise some

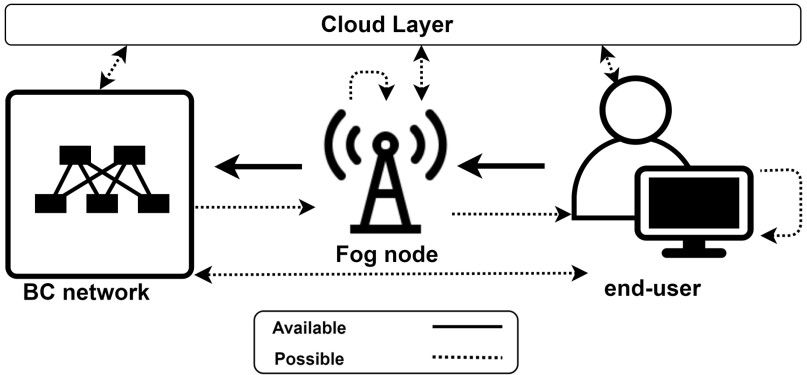

**Figure 6 Possible data flow schemes in an integrated Fog-BC system.**

error regarding the maximum allowed memory use by the interpreter. To solve this, one can modify the maximum limit using the sys.setrecursionlimit in the main function.

### Merkle trees

An MT, or a Hash Tree, is a data structure which is mostly a binary tree, whose leaves are chunks of data. Sub-consequently, each leaf is double hashed with its siblings to produce their new parent, which represents its two children. Hashes are recursively hashed together, in a binary manner, until obtaining one root that represents the whole tree. MTs are used in BCs such as Bitcoin to decrease the probability of security attacks, along with other security measures, to reach a level where it is (a) easy for light-weight nodes to validate new TXs and (b) computationally impractical to attack/alter a BC. That is, each TX in any given block is hashed with the next, and so on, so that one root hash of all TXs is saved in the block header. Using this root hash, and other components of the block, the hash of the block is generated. This means that not only a confirmed block is impossible to alter, but also a confirmed TX within a confirmed block.

However, not all BC systems deploy an MT approach due to some probable conflicts with system requirements or objectives. Thus, we decided to leave this to be implemented by developers according to the systems that need to be simulated, and we decided that the default configuration of BC nodes in the current version of FoBSim is to make all miners full node miners. That is, every miner locally stores a complete copy of the chain so that any TX can be validated according to TXs recorded locally. Additionally, there are different deployment models of MT approaches in different BC systems. That is, some BCs may deploy MTs for hashing other chunks of data/tokens instead of TXs.

To implement an MT approach in FoBSim, one can add a function that performs a loop through all TXs in a newly minted block, up to the last TX. After that, the root of the MT is added to the block before it is broadcast to the BC and the hash of the block is computed accordingly. Miners who receive a new block shall, accordingly, validate the added root. Hence, a validation step, to test the correctness of the MT root compared with TXs within the new block, needs to be added to the validation function in the miner

module of FoBSim. To make use of such added property, one can define a light-weight miner type which saves only the header of a newly confirmed block instead of the whole block. Accordingly, such type of miners validate new TXs according to this light chain of headers, hence consume less time, energy, and storage to maintain the work of the BC system.

### Digital signatures

As our main aim is to generally simulate TX generation, validation, and confirmation, in different BC-based, inter-operation, and consensus scenarios, we did not target security issues. This is because such issues are determined individually for each case to be simulated, leading to different mining economics. The security techniques and approaches in BC-based fog and IoT systems had been discussed in many previous works, such as *Sodhro et al. (2020)*. Specifically, digitally signed coins/tokens are primarily used in real-world applications of cryptocurrencies in order to prevent security attacks, such as the double spend attack. Different BC-based cryptocurrency systems use different mechanisms and protocols regarding signing and minting new coins, hence, different scenarios require the implementation of the reference coins and digital signing techniques to be simulated. Examples might include a research work that aims at comparing different signing protocols in different CAs. This being said, FoBSim does not target a specific cryptocurrency system, such as Bitcoin, yet it provides the generalized environment used in such systems, where problems and solutions can be implemented and emulated by researchers.

What the default version of FoBSim provides, however, is a simplified protocol of coin transfer between users. That is, each miner holds a locally saved record of user wallets, which is used in TX validation in case of payment BC functionality. We found that this approach can output similar results to those output by systems with signed coins, except that this approach allows a double spend attack in case of malicious end-users. If a scenario to be simulated, where there are some faulty/malicious entities among system users (which is not implemented in the default version of FoBSim), then digitally signed coins need to be implemented as well. Additionally, miner nodes in FoBSim are assumed to be trusted to send reports of confirmed blocks. Thus, reports sent by miner nodes to the network aiming to participate in voting regarding winning miners are assumed always legitimate. To sum up, FoBSim miners can track who, paid whom, how much, and they are trusted to participate in voting without a cryptographic proof. While, in other implementation approaches, FoBSim miners may track who has transferred, what units, of which stocks (i.e. digitally signed coins/tokens), to whom, and their votes regarding winning miners must be verified by network entities (i.e., by also adding the new block to their local chains, and following this addition with other new blocks, each newly added block can be considered, in a sense, a confirmation). Similarly, end-users who generate new TXs do not need to sign their generated TXs as they are assumed trusted (i.e. the default implementation of FoBSim does not include malicious end-users).

### Mining pools

Pool mining is the collaboration between miners to form mining pools and to distribute the earned rewards in accordance with pool policies to earn a steady income per miner (*Majeed, Kim & Hong, 2020*). Examples of such mining pools include BTC.com, F2Pool, and Slush Pool. Mining pools provide the advantages of making mining profits more predictable to miners and allowing small miners to participate. However, the existence of pool mining increases the probability of system centralization and discourages full nodes. The necessity of adding a mining pool extension to FoBSim is dependant on the scenario to be simulated. As the general idea of mining pools is to allow miners to perform mining under the umbrella of a named group, if one of the group miners finds a block, the award is divided among all group members according to the computational power each member provides. A mining pool is managed by a pool manager, whose protocol is defined according to the business model of the pool.

In the current version of FoBSim, all miners are full node miners. That is, each miner attempts to solve the puzzle using its own resources, to validate newly generated TXs and to accumulate them into new blocks. When a block is received by a full node, it is validated and confirmed locally (all miners save the whole BC for validation, verification, and confirmation). Consequently, any profits and awards, obtained because of the full miner work, are directly added to the miner's wallet. On the contrary, a miner receives an award that is proportional to the computational power it provides, even if it was the one who found the next block.

## CASE STUDIES

Following the validation and verification methods of simulation models presented in *Sargent (2013)*, we have so far discussed the technologies and the paradigms lying within our proposed FoBSim environment. Further, we highlighted our proposal novelty compared to other related works, discussed the event validity in FoBSim, and presented the algorithms and modules lying within to facilitate a structured walk-through validation.

Next, we follow an operational validity approach by presenting case studies that we simulated using FoBSim. The setup and behavior of FoBSim is discussed, and the results of the simulation runs are presented afterwards.

### Case 1: Comparing time consumption of PoW, PoS, and PoA

When we compare PoW, PoS and PoA in terms of average time consumed for block confirmation, PoW is expected to present the highest time consumption. This is because of the mathematical puzzle that each minter needs to solve in order to prove its illegibility to mint the next block. In PoS, on the other hand, the network algorithm randomly chooses the next minter, while it (slightly) prefers a miner with a higher amount of staked coins. Once a minter is chosen, all miners are informed about the generator of the next block and, thus, the minter needs to perform no tasks other than accumulating TXs in a new standard block. Other miners then accept the new block if it was generated by the minter they were informed about, hence the verification process takes nearly no time (assuming that the transmission delay between miners is set to 0). In simple versions of

**Table 3 Simulation parameters configuration for Case 1.**

| Simulation parameter\ Consensus | PoW | PoS | PoA |
| --- | --- | --- | --- |
| No. of miners | 5–500 | 5–500 | 5–500 |
| No. of neighbors per miner | 4 | 4 | 4 |
| Puzzle difficulty | 5–20 | – | – |
| Authorized miners | All | Random choice | 2–25 |
| Initial wallet | – | 1,000 | – |
| BC functionality | Data Management | Data Management | Data Management |
| BC deployment | end-user layer | end-user layer | end-user layer |

those two algorithms, all miners have the same source code, thus all miners may be minters, verifiers, and chain maintainers.

The PoA algorithm is the tricky one though. This is because all authorized miners mint new blocks, verify newly minted blocks, and maintain the chain locally. Meanwhile, other BC nodes verify new blocks and maintain the chain, but do not mint new blocks (*De Angelis et al., 2018*). Consequently, every BC node has a list of authorized entities, including the methods to verify their newly minted blocks. This implies that the more authorized entities, the more complex the verification can be on the receiver side. Accordingly, it is advised that a small number of entities be given authorization for decreasing the complexity of verification (*Binance Academy, 2020*). Meanwhile, the more maintainers in a PoA-based BC, the higher the overall security level of the system.

In this case study, we run FoBSim several times, with which we deploy different CAs under similar conditions. The simulation runs targeted specifically the measurement of the average time consumed by each CA, from the moment where a miner is triggered to mint a new block, until the minted block by this miner is confirmed by, at least, 51% of other BC miners. To accurately measure this average, we added some variables holding the starting time and the elapsed time, exactly before calling the build_block() function and right after a block is confirmed by reaching the required number of confirmations.

As described in Table 3, we changed the difficulty of the puzzle during the PoW-based BC simulation runs from an easy level (5), to a harder level (10), and finally to very hard levels (15) and (20). During the runs where PoA was used, we changed the number of authorized miners from 2/5 (2 authorized out of a total of 5 miners), 5/10, 10/20, and 25 authorized miners for the rest of runs.

As we wanted to abstractly measure the average confirmation time, we avoided the computational services and payment functionality, because both imply extra time consumption for performing the computational tasks, and validating the payments, respectively. We also avoided the identity management functionality because the number of TXs per end-user is limited by the number of ID attributes required to be saved on the chain. Hence, our best choice was the data management functionality. We kept the total number of TXs delivered to the mempool unchanged, which gives equivalent input for all simulation runs. However, we changed the number of TXs generated by each user as to be equal to the number of miners in each run. More precisely, as the total

**Table 4 Results of Case-1, where the PoW puzzle difficulty ranges from 5 to 20, and the number of Miners (M) ranges from 5 to 500.**

|  | M = 5 | M = 10 | M = 20 | M = 50 | M = 100 | M = 500 |
|---|---|---|---|---|---|---|
| PoS algorithm | 0.018 | 0.06 | 0.18 | 0.046 | 0.09 | 0.19 |
| PoA algorithm | 0.002 | 0.008 | 0.03 | 0.2 | 0.41 | 2.94 |
| PoW-5 algorithm | 0.08 | 0.36 | 2.1 | 1.31 | 6.15 | 60.6 |
| PoW-10 algorithm | 0.07 | 0.44 | 2.1 | 2.03 | 5.21 | 58.9 |
| PoW-15 algorithm | 0.25 | 0.42 | 2.23 | 2.26 | 6.18 | 74.76 |
| PoW-20 algorithm | 6.02 | 9.5 | 24.2 | 59.62 | – | – |

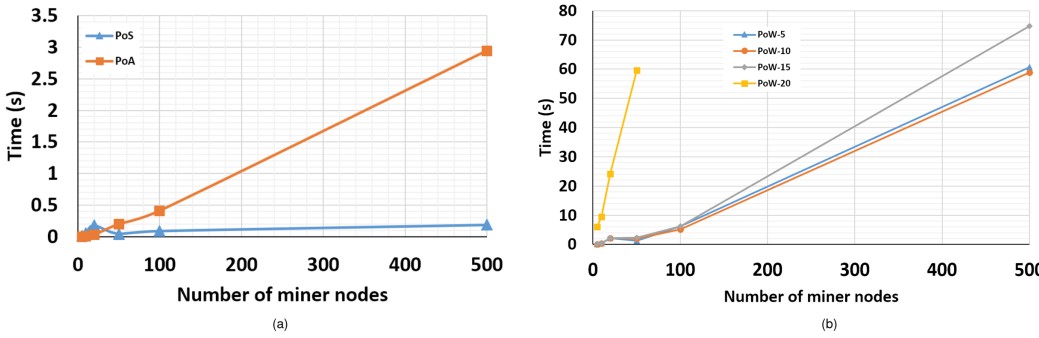

(a)                                          (b)

**Figure 7 Average block confirmation time (A) consumed by PoS-based BC vs. PoA-based BC, relatively to the number of miner nodes (B) consumed by PoW-based BC (the cases of difficulty = 5, 10, 15, and 20), relatively to the number of miner nodes.**

number of TXs is determined using Eq. (2), where *a*, *b* and *c* are the number of fog nodes, the number of end-users, and the number of TXs per end-user, respectively, the values of those variables fluctuated in each run. Concerning the runs where a PoS is deployed, miner nodes were initiated with a wallet that has 1,000 coins, allowing miners to stake random amounts of coins. Additionally, winning miners were awarded 5 coins for each confirmed block they had minted.

$$|TXs| = a \times b \times c \tag{2}$$

We deployed the FoBSim environment on Google Cloud Platform, using a C2-standard-16 (up to 3.8 GHz, 16 vCPUs, 64 GB memory), with Debian OS. We have chosen to place the BC in the end-user layer for all runs, not for any reason other than testing the reliability and stability, of FoBSim components and results, in such complex inter-operable (*Belchior et al., 2020*) Edge-Fog-BC scenarios. Table 4 presents the exact results we obtained, which are depicted in Figs. 7A and 7B.

According to the results obtained from the simulation runs, one can notice that PoW-based BCs consume much more time to confirm a block, than PoA- and PoS-based BCs, which is inline with the theoretical and experimental results of most previous research. Additionally, the average block confirmation time, in PoW-based and PoA-based BCs, seems to be directly proportional to the BC network size, which complies with the

**Table 5 Simulation parameters configuration for Case-2, where the Gossiping property is interchangeably activated and deactivated.**

| Simulation parameter | Puzzle difficulty effect | Transmission delay effect |
|---|---|---|
| No. of fog nodes | 5 | 5 |
| No. of users per fog node | 5 | 5 |
| No. of TX per user | 5 | 5 |
| No. of miners | 100 | 100 |
| No. of neighbors per miner | 2 | 2 |
| No. of TX per block | 5 | 5 |
| Puzzle difficulty | 5, 10, 15, 20 | 20 |
| Max end-user payment | 100 | 100 |
| Miners' initial wallet value | 100 | 100 |
| Mining award | 5 | 5 |
| Delay between neighbors | 0 | 0, 5, 10, 15, 20 |

results recently presented in *Misic, Misic & Chang (2020)*. Comparatively, an average block confirmation time in a PoS-based BC seems unaffected by the network size, which complies with the expectations recently presented in *Cao et al. (2020)*.

## Case 2: Capturing the effect of using the gossip protocol

In this case, we compare the number of chain forks at the end of several simulation runs, where we interchangeably activate and deactivate the gossiping property in a PoW-based BC. Accordingly, one can notice the effect of gossiping on ledger finality under different conditions, namely the puzzle difficulty and the transmission delay between miners. As it was mentioned in "Gossip Protocol", gossiping is a continuous process during the life time of the network, which implies that miners would mostly have different chain versions at any given moment. In this case, we detect the number of chain versions at the end of simulation runs, which can be decreased to one version under strictly designed parameters, such as medium network size, high puzzle difficulty, low transmission delay, low number of neighbors per miner, etc. Nevertheless, our goal in this case is to demonstrate how the activation of the gossiping property during a simulation run on FoBSim can decrease the number of chain versions and, thus, it can positively contribute to the consistency of the distributed ledger. For this case, we also deployed the FoBSim environment on the Google Cloud Platform, using a C2-standard-16 VM (up to 3.8 GHz, 16 vCPUs, 64 GB memory), with Ubuntu OS.

Table 5 presents the initial configuration in each simulation scenario, while Tables 6 and 7 present the results we obtained by running the described scenarios, which are depicted in Figs. 8A and 8B. As can be noted from the results, the default gossip protocol in FoBSim could decrease the number of chain versions at the end of each simulation run. Although the number of chain versions did not reach the optimum value (i.e., one chain version), it is obvious that activating the gossiping property decreases the number of

**Table 6 Results of Case-2, where the puzzle difficulty ranges from 5–20, and the Gossiping in FoBSim was interchangeably activated and deactivated.**

| Configuration | diff. = 5 | diff. = 10 | diff. = 15 | diff. = 20 |
|---|---|---|---|---|
| Gossip activated | 81 | 70 | 57 | 16 |
| Gossip deactivated | 92 | 98 | 100 | 67 |

**Table 7 Results of Case-2, where the transmission delay between neighbors ranged from 0–25 ms., and the Gossiping in FoBSim was interchangeably activated and deactivated.**

| Configuration | T.D. = 0 | T.D. = 5 | T.D. = 10 | T.D. = 15 | T.D. = 25 |
|---|---|---|---|---|---|
| Gossip activated | 12 | 18 | 14 | 26 | 33 |
| Gossip deactivated | 15 | 39 | 59 | 68 | 76 |

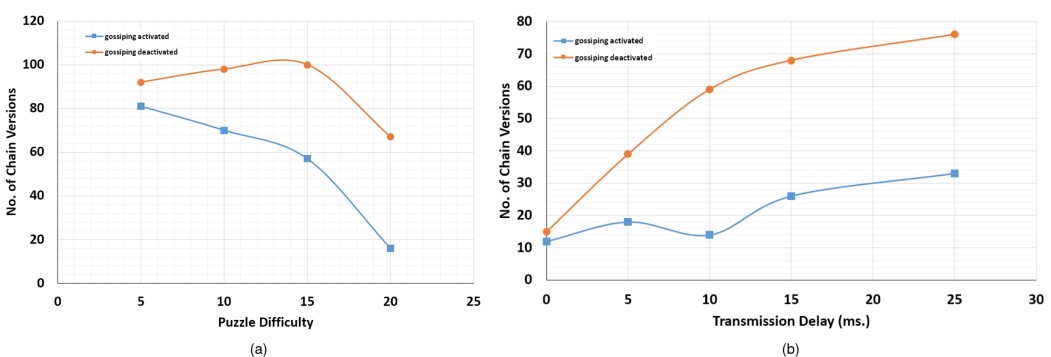

**Figure 8 The effect of activating the gossiping protocol in FoBSim, on the number of chain versions at the end of PoW-based BC simulation runs, where (A) the puzzle difficulty fluctuates from 5 to 20 and (B) the transmission delay between neighboring miners fluctuates from 0 to 25 ms.**

chain versions at each simulation run and, thus, enhances the distributed ledger consistency.

## Case 3: Comparing deployment efficiency of BC in the fog layer vs. end-user layer

In this case, we compare BC deployment efficiency in the fog layer and end-user layer. The efficiency we are seeking is determined by both the total time needed to perform all requested BC services and total storage cost. That is, less time and storage needed to perform all tasks (e.g., confirm all newly minted blocks or run the generated SCs) indicates higher efficiency of the BC system. To fairly compare the BC efficiency when deployed in those two layers, we stabilize all BC parameters that are configurable in FoBSim, except for the number of miner nodes to deduce the trend in total time consumption when the network dynamically allows for new nodes to join the network. We deployed the FoBSim tool on the Google Cloud Platform, using a C2-standard-16 VM (up to 3.8 GHz,

**Table 8 Simulation parameters configuration for Case-3, where the efficiency of BC is assessed in the fog layer and end-user layer, in terms of total run time and total storage cost.**

| Simulation parameter | For total time efficiency | For total storage efficiency |
| --- | --- | --- |
| No. of fog nodes | 10–100 | 100 |
| No. of users per fog node | 2 | 5 |
| No. of TX per user | 2 | 5 |
| No. of miners | 10–100 | 100 |
| No. of neighbors per miner | 3 | 5 |
| No. of TX per block | 5 | 5 |
| Puzzle difficulty | 20 | 15 |
| Max end-user payment | 100 | 100 |
| Miners' initial wallet value | 1,000 | 1,000 |
| Mining award | 5 | 5 |
| Delay between neighbors | fog layer: 12 ms., end-user layer: 1,000 ms | fog layer: 12 ms., end-user layer: 1,000 ms |

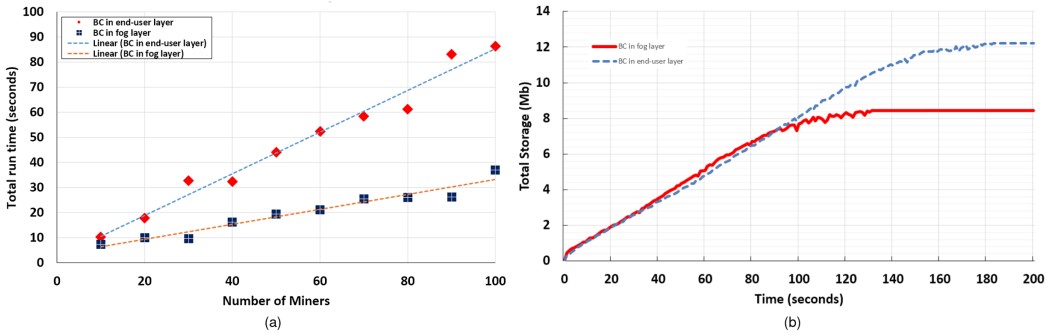

**Figure 9 BC efficiency comparison while deployed in end-user layer vs. fog layer, in terms of (A) total elapsed time for the BC network to perform requested services, and (B) total storage used by the BC network to perform requested services.**

16 vCPUs, 64 GB memory), with Ubuntu OS. The detailed parameter configuration while running the described scenarios is presented in Table 8.

Recalling the results presented in *Bi, Yang & Zheng (2018)* and *Li et al. (2017),* average transmission delay between miners in the fog layer can be estimated by 12 ms., while it can be estimated between miners in the end-user layer to 1,000 ms. (higher transmission delays were reported in well known BC networks, such as Bitcoin, in *Sallal (2018)*). We simulated the data management BC service and PoW consensus with gossiping activated. According to Eq. (2), the number of requested tasks was automatically modified due to the continuous change in the number of fog nodes (since we oscillated the number of fog nodes to deduce the trend of total time consumption). The total average time for performing requested BC services, in similar simulation sittings, while the BC is deployed in end-user and fog layers, is compared in Fig. 9A.

To accurately measure the storage cost during the simulation run, we implemented an independent Python code, available in the FoBSim repository, namely storage_cost_analysis.

py. As described in "FoBSim Modules", the output analysis files, ledgers, wallets, etc. of running a given simulation scenario using FoBSim, are automatically saved in a folder titled "temporary" within the same repository. Thus, our implemented storage analyzer aims at regularly (i.e., every one second as a default sitting) measuring the size of this temporary folder while the simulation is running. The measured sizes are then saved into an Excel sheet to facilitate performing the analysis we are seeking. To exemplify this, the total storage used by the BC network is compared in Fig. 9B, where similar simulation sittings were configured (detailed in Table 8), except for the layer where the BC is deployed.

It can be noted from the results presented in the third case that deploying the BC network in the fog layer may enhance its efficiency in terms of total time consumed to perform similar tasks in similar configuration, and in terms of total storage cost by the BC network to maintain the same distributed ledger (same number of confirmed blocks by the end of the simulation run).

## CONCLUSIONS

In this paper, we proposed a novel simulation tool called FobSim that mimics the interaction between the entities of an integrated Fog-Blockchain system. We briefly described the architectural elements of Fog Computing (FC) and Blockchain (BC) technologies, and designed FoBSim in order to cover all the elements we described. We deployed three different consensus algorithms, namely PoW, PoS and PoA, and different deployment options of the BC in an FC architecture, namely the end-user layer and the fog layer. Additionally, we fine tuned the FoBSim modules so that various services, provided by FC and BC, can be adopted for any proposed integration scenario.
The services that can be simulated are distributed payment services, distributed identity services, distributed data storage and distributed computational services (through Smart Contracts). In our paper, we described the modules of FoBSim, the TX modeling, the Genesis block generation, the gossiping in FoBSim, the Consensus Algorithms, TX and block validation, incentive mechanisms, and other FoBSim strategies. We validated FoBSim with case studies: the first compares the average time consumption for block confirmation in different consensus algorithms, while the second analyzes the effect of gossiping on the consistency of the distributed ledger, in fluctuated puzzle difficulty and transmission delay configurations. The last case compared the efficiency of the BC network, in terms of total time consumption and total storage required to perform similar tasks, when deployed in the fog layer against the end-user layer. The results of the first case showed that the PoS algorithm provides the least average block confirmation time, followed by PoA and PoW, respectively. The results of the second case showed how the gossiping protocol, implemented within FoBSim, effectively contributes to enhance the consistency of the distributed ledger. The last case showed that deploying the BC network in the fog layer may drastically enhance the BC performance, in terms of total execution time and total storage cost, due to low transmission delay between miners.

In the future releases of FoBSim, we are willing to make more CAs available, as well as enhancing the identity management scheme in FoBSim. We will further investigate

adding the reputation management service in a generalized and simple manner so that analysis can be provided, while proposed reputation management ideas, conditions, or properties can be easily implemented/modified.

### Funding

This research was supported by the Hungarian Scientific Research Fund under the grant number OTKA FK 131793, by the grant NKFIH-1279-2/2020 of the Ministry for Innovation and Technology, Hungary, and by the National Research, Development and Innovation Office within the framework of the Artificial Intelligence National Laboratory Programme. The funders had no role in study design, data collection and analysis, decision to publish, or preparation of the manuscript.

### Grant Disclosures

The following grant information was disclosed by the authors:
Hungarian Scientific Research Fund: OTKA FK 131793.
Ministry for Innovation and Technology, Hungary: NKFIH-1279-2/2020.

### Competing Interests

The authors declare that they have no competing interests.

### Author Contributions

- Hamza Baniata conceived and designed the experiments, performed the experiments, analyzed the data, performed the computation work, prepared figures and/or tables, authored or reviewed drafts of the paper, and approved the final draft.
- Attila Kertesz conceived and designed the approach and the experiments, analyzed the data, authored and reviewed drafts of the paper, and approved the final submission.

### Data Availability

The source code of the simulator is available at GitHub: https://github.com/sed-szeged/FobSim.

### Supplemental Information

Supplemental material for this article can be found online at http://dx.doi.org/10.7717/peerj-cs.431#supplemental-information.

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
