# Peer review of "FoBSim: an extensible open-source simulation tool for integrated fog-blockchain systems"

_PeerJ Computer Science, doi:10.7717/peerj-cs.431_

## Round 0.1 · original submission · Major Revisions

Dear authors,

In general, the reviewers state that this is very interesting work and surely worth publication, but some aspects should be improved. For this, please take into account the comments by the two reviewers. Both reviewers state that the evaluation/experimentation could be further extended, so please take this especially into account.

Best regards,

Stefan Schulte

Reviewer 1 ·

Basic reporting

1) The writing of the paper should be improved for example
Section 4.4 Them selves.

Experimental design

2) It is said that the complex tasks are forwarded to the cloud. However, I don’t see any explanation for these kind of tasks and how are they linked to the blockchain network? It needs explanation in the paper.

3) “Meanwhile, the communications within the BC network and with the fog layer are configurable”. What exactly this statement signifies. What exactly is the definition of communication here?

4) It is said that the blockchain can be placed in Fog-layer as well as End user layer. But how do we decide this? Is there a possibility that BC layer act independently in the simulator, i.e., BC should not be placed in Fog-layer and End user layer?

5) Similarly, it is said that the BC can be implemented in cloud? Is it reasonable to put BC nodes in cloud architecture?

Validity of the findings

6) There is an initiate_miner() function. If BC functionality is implemented in End-users, all the nodes will act as a miner or there is a possibility that few nodes act as a miner and few nodes functions normally? If that is the case, can we tune the number of participating nodes in the blockchain network?

7) Finally, along with the time consumption analysis, authors should also perform the storage cost analysis for the transactions in FoBSim.

Additional comments

Opinion:
Minor Revision
Summary:
To validate an integrated Fog-Blockchain protocol or method implementation, before the deployment phase, a suitable and accurate simulation environment is needed. Such validation should save a great deal of costs and efforts on researchers and companies adopting this integration. Current available simulation environments facilitate Fog simulation, or BC simulation, but not both. In this paper, authors introduce a Fog-Blockchain simulator, namely FoBSim, with the main goal is to ease the experimentation and validation of integrated Fog-Blockchain approaches. According to the proposed workflow of simulation, different Consensus Algorithms (CA), different deployment options of the BC in the FC architecture, and different functionalities of the BC in the simulation are implemented. Furthermore, technical details and algorithms on the simulated integration are provided.
Strong Points
1) The paper is well presented and organized. Moreover, the novelty of the work is presented correctly.
2) The paper presents extensive and tabular comparison with state-of-the-art.
3) The paper is technically sound and authors have also tried to explain the possible problems associated with their proposed design.

Weak Points
1) The writing of the paper can be improved further.
2) Few concepts and statements in the paper are confusing. They should be explained further.
3) The experimental analysis of the simulator can be presented better.

Conclusion
To conclude, the paper is well presented and technically sound. Almost all the concepts are explained well however, the paper can be improved further by addressing the above-mentioned comments.

Reviewer 2 ·

Basic reporting

Well-written, easy to follow.

Experimental design

Does not evaluate different fog computing infrastructures, geo-distribution, layers, etc.

I think this is the selling point of fog computing, and I would expect experiments that show the effect of fog computing on block chain applications. However, the authors only evaluate block chain applications in a fixed "fog" infrastructure, which may leave the readers a bit disappointed.

I'd suggest to add also experiments where a given blockchain application is tested how it behaves in different fog computing infrastructures. This would shed light on the issue whether a given blockchain application would perform well in different fog settings.

Validity of the findings

Findings seem valid, but are very limited.

Additional comments

While the article is well-written and provides a lot of details of the simulation system proposed, there are some fundamental questions that are not answered.

- Motivation:
Why is fog computing and blockchain a good match

- Blockchain technology:
Important concepts are left unexplained. For instance, I miss an explanation of private vs. public blockchains. The authors seem only to address public blockchains; but they also make references to private blockchains like Hyperledger Fabric.

- Evaluations without real application:
There are no real blockchain applications evaluated, only consensus protocols. It stays unclear how real applications would benefit (if at all) from fog computing. Further, as mentioned above, different fog infrastructures are not compared to each other, which would be interesing.

The authors use too many abbreviations, like BC, FC, CA, SC, ... which make the text sometimes hard to comprehend.

---

## Round 0.2 · accepted · Accept

I have also read the paper and would like to provide you with my proof-read version on Wednesday (I am currently in home office and need a scanner ;)). For now, apart from the comments of Reviewer 2, please think about the following:
* It may make sense to use line numbers in the algorithms and use these to explain the algorithms in more detail.
* Please use vector graphics in order to improve readability and scalability of your figures.
* As Reviewer 2 points out, it may make sense to integrate some figures from the appendix into the main part. Since PeerJ CS papers do not have a page limit, this is actually a good idea.

Reviewer 1 ·

Basic reporting

The paper has addressed all previous comments.

Experimental design

The paper has addressed all previous comments.

Validity of the findings

The paper has addressed all previous comments.

Additional comments

The paper has addressed all previous comments.

Reviewer 2 ·

Basic reporting

Typo in Figure 1 in the appendix: "Chose" --> "Choose"

In Figure 1, two states are named "Start the simulation". Are there two simulations?

It seems some figures in the appendix / supplemental material are fundamental to the paper. They are heavily referred to in the text. It feels awkward to have them in the appendix. To me, an appendix is something optional for the interested reader, and the reader should be able to fully follow the paper without looking into the appendix.

If you want to save space, there are other possibilities than moving important figures to the supplemental material-

Experimental design

The additional experiment is good (Use case 3).

Validity of the findings

no comment

Additional comments

I'm fine with the revision, most of my comments have been addressed.

However, the question of what material goes to the main paper and what goes to the appendix should be reevaluated.